# Study on the Role of Phytohormones in Resistance to Watermelon *Fusarium* Wilt

**DOI:** 10.3390/plants11020156

**Published:** 2022-01-07

**Authors:** Feiying Zhu, Zhiwei Wang, Yong Fang, Jianhua Tong, Jing Xiang, Kankan Yang, Ruozhong Wang

**Affiliations:** 1Hunan Provincial Key Laboratory of Phytohormones, College of Bioscience and Biotechnology, Hunan Agricultural University, Changsha 410128, China; feiyingzhu@hunaas.cn (F.Z.); tongjh0421@hunau.edu.cn (J.T.); XJ459970488@163.com (J.X.); 2Hunan Agricultural Biotechnology Research Institute, Hunan Academy of Agricultural Sciences, Changsha 410125, China; yongfang@hunaas.cn; 3Institute of Biotechnology, Hunan Academy of Agricultural Sciences, Changsha 410125, China; wangzhiwei119@163.com (Z.W.); kankankkyang@163.com (K.Y.)

**Keywords:** salicylic acid, jasmonic acid, abscisic acid, watermelon, *Fusarium* wilt, resistance

## Abstract

*Fusarium* wilt disease is one of the major diseases causing a decline in watermelon yield and quality. Researches have informed that phytohormones play essential roles in regulating plants growth, development, and stress defendants. However, the molecular mechanism of salicylic acid (SA), jasmonic acid (JA), and abscisic acid (ABA) in resistance to watermelon *Fusarium* wilt remains unknown. In this experiment, we established the SA, JA, and ABA determination system in watermelon roots, and analyzed their roles in against watermelon *Fusarium* wilt compared to the resistant and susceptible varieties using transcriptome sequencing and RT-qPCR. Our results revealed that the up-regulated expression of *Cla97C09G174770*, *Cla97C05G089520*, *Cla97C05G081210*, *Cla97C04G071000*, and *Cla97C10G198890* genes in resistant variety were key factors against (*Fusarium oxysporum f. sp. Niveum*) FON infection at 7 dpi. Additionally, there might be crosstalk between SA, JA, and ABA, caused by those differentially expressed (non-pathogen-related) NPRs, (Jasmonate-resistant) JAR, and (Pyrabactin resistance 1-like) PYLs genes, to trigger the plant immune system against FON infection. Overall, our results provide a theoretical basis for watermelon resistance breeding, in which phytohormones participate.

## 1. Introduction

Watermelon (*Citrullus lanatus*) *Fusarium* wilt disease pose a serious threat to watermelon quality and yield [1,2,3]. Different techniques, including chemical control [4], biological control [5,6], grafting [7], and the use of disease-resistant cultivars [8], are utilized to overcome this disease. Many researchers have focused on recognition competition between the host plants, pathogens, pathogenic factors, and host plant defense factors [9,10,11]. Phytohormones were reported as signaling molecules acting in plant communication, which play important roles in plant growth and stress responses [11,12,13]. For example, salicylic acid (SA) has an essential role in systemic acquired resistance (SAR) [14] and is also involved in shaping the plant microbiomes to increase the plant immune capacity [15]. Jasmonic acid (JA) is recognized as another major defense hormone, which identified crosstalk between SA pathways [16,17]. Resent discoveries reported the important role of abscisic acid (ABA) to abiotic stress, and it has emerged as a modulator of the plant immune signaling network [18,19]. Similarly, most studies have demonstrated that SA synthesis is induced under stress conditions in watermelon. For instance, Zhu comparatively analyzed two contrasting watermelon genotypes during fruit development and ripening based on transcriptome, and suggested that ABA and ethylene might equally contribute to regulating watermelon fruit quality [20]. Xu’s research has noticed that JA biosynthesis genes were induced and activated at the early stage of (*Fusarium oxysporum f. sp. Niveum*) FON infection in watermelon cultivated accompanying by wheat [21]. Cheng found out that low temperature induced SA production, and it might cooperate with redox signaling to regulate watermelon resistance [22]. Moreover, Guang recently identified that the exogenous JA, SA, and ET treatment significantly up-regulated the CIOPR gene under root knot nematode infection [23]. Therefore, we hypothesized that phytohormones may play a crosstalk role in triggering the watermelon plant immune system.

With the rapid development of molecular biology, some new biotechnology methods have been widely used in watermelon plant breeding, as well as disease resistance [23,24,25]. For instance, Guo has identified the genome sketch of watermelon material 97103 and analyzed its genomics, which has paved the way for botanists to study watermelon at molecular level [24]. Li has identified many disease resistance-related candidate genes during the infection process of watermelon fruit against Cucumber Green Mottle Mosaic Virus (CGMMV) by RNA Seq, which lay a foundation for further gene functional study [25]. However, the molecular mechanism of phytohormones network relationships in watermelon plants was less known. Therefore, in this experiment, our aim is to explore the essential role of SA, JA, and ABA in watermelon resistance to *Fusarium* wilt disease by comparing differences in their concentration and signal-related gene expression in resistant and susceptible varieties. These studies will provide a comprehensive resource for identify the genes associated with the phytohormones of watermelon resistance breeding.

## 2. Results

### 2.1. Comparison Analysis of Phenotype and Fusarium Wilt Disease Occurrence in Resistant and Susceptible Watermelon after FON Inoculation

In order to clarify the effect of different watermelon varieties on *Fusarium* disease resistance, the phenotypes of watermelon seedlings and the disease incidences after pathogen infection were explored. The trial crops were watermelon resistance cultivated variety PI296341 and susceptible cultivated variety, zaojia 8424. The watermelon seedling nutrition bowl was cultivated and grown in a biochemical incubator at Tm 25 °C, light 16 h/Tm 18 °C, and dark 8 h. When the watermelon seedling growth stage was at two leaves apart, aliquots of 10^6^ conidia/mL FON were added into the root zone of each watermelon plant. Samples referred to as S7, Susceptible cultivar + mock-inoculation control (H_2_O_2_), 7 dpi (7 days post inoculation); R7, Resistant cultivar + mock-inoculation control (H_2_O_2_), 7 dpi (7 days post inoculation); SF7, Susceptible cultivar + FON, 7 dpi (7 days post inoculation); and RF7, Resistant cultivar + FON, 7 dpi (7 days post inoculation). The results showed that the resistant watermelon seedlings grew bigger with more leaves without any disease symptoms, while the susceptible variety plants had obvious disease symptoms such as yellowing and wilting at 7 days post inoculation (Figure 1A). Comparing analysis of the root’s phenotypic changes after FON infection shows more fibrous roots in the resistant group than that of susceptible one (Figure 1B). The fresh weight of roots in the resistant cultivar was nearly three times heavier than that of the susceptible cultivar (Figure 2A). Furthermore, the disease incidence of watermelon *Fusarium* wilt in SF7 was 33.3%, while it was 0% in RF7 (Figure 2B). The question is, how the resistant variety was able to control the disease.

### 2.2. Dynamic Changes in MDA Content and PAL, POD Enzyme Activities after FON Inoculation

We compared the dynamic changes in malondialdehyde (MDA) content, phylalnine ammonialyase (PAL) enzyme activity, and peroxidase (POD) enzyme activity of resistant and susceptible watermelon roots at different stages after FON infection. The results showed that there was a continuous increasing of MDA content in susceptible watermelon roots, which represented the changes in plant health condition at different stages after FON infection (Figure 3A). After FON treatment, the POD enzyme activity in RF group was increased significantly from 12 hpi to 1 dpi, and then decreased continuously. Notably, the POD enzyme activity in SF was nearly two times higher than that in RF at 3 dpi (Figure 3B). Our results showed that the PAL enzyme activity in RF was nearly twice as high as SF at 3 dpi, but then decreased rapidly, up to only half that compared with the SF group at 7 dpi (Figure 3C). Our results indicate that the POD and PAL may have important roles as resistant varieties against FON infection at an early stage.

### 2.3. Comparison Analysis of SA, JA, and ABA Contents at 7 dpi

In order to explore the physiological mechanism of phytohormones in against watermelon *Fusarium* wilt while the symptoms appeared (e.g., rotted, discolored), we established a determination system for the SA, JA, and ABA content in watermelon roots, as shown in Appendix A. Here, we comparison analyzed the SA, JA, and ABA content in resistant and susceptible watermelon roots at 7 dpi. The results showed that the SA content in the susceptible group was significantly higher than that in the resistant group, but the content of JA and ABA had no significant difference (Figure 4).

### 2.4. Comparison Analysis of Transcriptome Differences at 7 dpi

To examine the molecular mechanism of *Fusarium* wilt resistance, we used transcriptome sequences to analyze the difference expressed genes between resistant and susceptible watermelon cultivars at 7 dpi as well.

The violin diagram showed that the covered distribution of gene expression levels was uniform in different samples after calculating the expression values (FPKM) of all genes (Figure 5A). In order to further compare the different gene expressions in resistant and susceptible varieties treated by pathogens, we conducted PCA analysis on the distribution of different genes, which showed differences between these two varieties, as seen in Figure 5B. The volcanic map indicates that there were 21,719 genes detected, with 6286 significantly differentially expressed genes compared, in which 2970 were up-expressed, and 3316 genes were down-expressed, between these two varieties (Figure 5C).

### 2.5. Functional Annotation of Genes Expressed

The scatter plot was used for comparison analysis of the genes’ functional enrichment in differential varieties. The KEGG enrichment significant analysis results indicate that most of the differently expressed genes were highly enriched in carbon metabolism, glutathione metabolism, biosynthesis of amino acids, glycolysis, pyruvate metabolism, and terpenoid backbone biosynthesis pathways (Appendix A). The up-regulated differently expressed genes enriched in arachidonic acid metabolism, spliceosome, and terpenoid backbone biosynthesis pathways (Appendix A). Furthermore, we identified that there were 10 DEGs related in terpenoid backbone biosynthesis pathways (Figure 6A,B). On the other hand, the GO enrichment significant analysis results indicate that most differential genes belong to functions such as response to stress and vesicle-mediated transport. The functions of significantly differently expressed genes were mainly in the Cytoplasmic vesicle part, vesicle, cytoplasmic vesicle, coated vesicle membrane, Golgi-associated vesicle membrane, cytoplasmic vesicle membrane, coated vesicle, vesicle coat, vesicle membrane, and Golgi-associated vesicle (Appendix A). The heatmap showed that there were 10 down-regulated genes (*Cla97C02G032030/Cla97C11G210770/Cla97C06G117130/Cla97C11G213000/Cla97C11G219070/Cla97C06G109940/Cla97C02G031650/Cla97C05G088840/Cla97C11G211210/Cla97C03G068230*) and one up-regulated gene (*Cla97C04G073730*) (Figure 6C). In addition, the up-regulated genes were mainly enriched in response to stress, nucleus, nucleoplasm part, nucleoplasm, nuclear lumen, mediator compels, intracellular organelle lumen, organelle lumen, and membrane-enclosed lumen (Figure 6D). In conclusion, the results laid a foundation for further study of the differential metabolic pathways in the process of watermelon resistance to *Fusarium* wilt.

### 2.6. Analysis of Hormone-Related DEGs

To examine the molecular mechanisms of genes involved in phytohormones signal transduction pathways, we used a heatmap to analyze the differences in resistant and susceptible watermelon varieties (Figure 7).

Most genes involved in the SA pathway were highly expressed in SF7 compared with RF7, where only *Cla97C01G009310* significantly up-regulated expression in RF7. There were 24 DEGs in the JA pathway, *Cla97C07G130430*, *Cla97C10G192210*, *Cla97C09G174730*, *Cla97C10G186220*, *Cla97C05G100240*, *Cla97C04G078620*, *Cla97C05G105650*, and *Cla97C05G100320,* all of which significantly up-regulated expression in RF7 compared with SF7. Among the 37 significantly DEGs involved in ABA pathway, there were 14 DEGs significantly up-regulated in the expression of RF7 compared with SF7, namely, *Cla97C06G123770*, *Cla97C09G174770*, *Cla97C10G186260*, *Cla97C01G023840*, *Cla97C11G221400*, *Cla97C08G158420*, *Cla97C01G010380*, *Cla97C10G188860*, *Cla97C09G172410*, *Cla97C03G063210*, *Cla97C01G020790*, *Cla97C05G106700*, *Cla97C07G134120*, and *Cla97C01G006610*. There were 48 significantly DEGs involved in the ET pathway, *Cla97C06G114420* and *Cla97C11G223860* highly up-regulated expression in RF7 compared with SF7; only *Cla97C05G108770* was expressed in RF7. There were 13 significant DEGs involved in the CTK pathway, *Cla97C11G207380* and *Cla97C05G099290* highly up-regulated expression in RF7 compared with SF7; only *Cla97C02G040700* was expressed in RF7. Among the 15 significantly DEGs involved in IAA pathway, *Cla97C08G155010* and *Cla97C11G217540* highly up-regulated expression in SF7 compared with RF7; only *Cla97C05G107110* was expressed in SF7. *Cla97C05G099610*, *Cla97C08G145860*, and *Cla97C05G099600* significantly up-regulated expression in RF7 compared with SF7 in the GA pathway (Appendix A).

### 2.7. Bioinformatics Analysis of Candidate Genes

In our study, a number of DEGs involved in SA, JA, and ABA pathways were identified at the stage where watermelon *Fusarium* wilt symptoms appeared (e.g., rotted, discolored), which was 7 days after FON inoculation. A schematic overview of DEGs related to different key components of SA, JA, and ABA signaling pathways are shown in Figure 8.

Notably, the expression of the CIPAL and BAH family genes were key to SA accumulation in watermelon plants. The interaction network of SA- (Figure 9A), JA- (Figure 9B), and ABA- (Figure 9C) related genes were analyzed. We observed that there were four NPRs, one JAR, and four PYLs genes significantly expressed compared between resistant and susceptible watermelon materials. The NPR5 may activate plant immune system through TGAs, while JAR1 has interaction with pathogen-related proteins (PRs) through Lipoxygenase. The three-dimensional structures of ABA receptor PYL, specific protein phosphatases type-2C (PP2Cs), and SAPK paves the way for ABA agonists to modulate the plant stress response.

### 2.8. Expression Verification of 10 DEGs

To further test the hypotheses about the different expressed genes in differed watermelon cultivar roots, we examined evidence from the RT-qPCR results (Appendix A and Figure 10).

The results indicated that the gene of *Cla97C04G073730* (clathrin light chain, cellular component) was significantly activated in two varieties after FON infection at 7dpi, and results confirmed the expression of PYL (*Cla97C09G174770*), PP2C (*Cla97C05G089520*), JAR1 (*Cla97C05G081210*), NPR (*Cla97C04G071000*, *Cla97C10G198890*), and BAH (*Cla97C04G071000*, *Cla97C10G198890*) genes had significant differences. The significant expression of NPR genes (*Cla97C04G071000*, *Cla97C10G198890*) suggests that they may have different functions in different watermelon cultivars to regulate plant defense systems.

### 2.9. Prediction Analysis of Phytohormones cis-Acting Regulatory Elements of 9 DEGs

The diverse expressions of JAR, NPRs, and PYLs family genes in watermelon suggest that they may play a synergistic role in regulating watermelon resistance to *Fusarium* wilt. Furthermore, we analyzed nine DEGs related to JAR, NPRs, and PYLs genes to identify their phytohormones cis-elements (Appendix A). Our results indicated that the NPR genes (*Cla97C01G009310*, *Cla97C07G137510*, and *Cla97C04G071000*) had SA, MeJA, IAA, or GA response elements but *Cla97C10G198890* only had ABA response elements. The PYL genes (*Cla97C05G081110*, *Cla97C09G172410*, and *Cla97C09G174770*) had ABA, SA, MeJA, IAA, or GA response elements, but *Cla97C10G186260* only had MeJA response elements. Moreover, the JAR1 (*Cla97C05G081210*) had ABA, SA, MeJA, IAA, and GA response elements.

## 3. Discussion

Watermelon (*Citrullus lanatus*) *Fusarium* wilt disease, caused by *Fusarium oxysporum f. sp. niveum* (FON), is a severe threat to watermelon yield and quality [1,2,3]. Symptoms such as discolored roots, and rotted and brown vascular bundles appear in *Fusarium* wilt-diseased watermelon plants [4]. Previously, scientists have discovered that phytohormones play important roles in watermelon plant growth and stress responses [20,21,22]. For instance, Ren reported that exogenously applied 100 μM of SA stimulates β-1,3-glucanase activity in watermelon leaves after FON inoculation [26]. Lv revealed signaling transduction nets between JA and SA during wound-induced agarwood production in A. sinensis [27]. Li has revealed the essential role of ABA in grape root restriction [28]. Therefore, in this experiment, we focused on exploring the essential role of SA, JA, and ABA in watermelon resistance to *Fusarium* wilt by comparing their concentration and signal-related gene expression differences in resistant and susceptible varieties.

Researchers have reported that the peroxidation of membrane lipids is activated when plants face environmental stresses [29]. PAL has been considered as the plant defense enzyme [30]. It is closely related to the synthesis of various secondary metabolites, such as lignin, isoflavone phytoalexin, and flavonoid pigment, which can contribute to disease resistance [31,32]. Similarly, our results indicate that the POD and PAL may have important roles in resistant variety against FON infection at an early stage (Figure 3). Our results also showed that the SA was induced after FON attack in both varieties of watermelon roots, but had a higher enrichment in the susceptible group at the onset stage (Figure 4). The results confirm that the low concentration of SA is conducive to plant disease resistance while excessive accumulation of SA may lead to plant death [11,13]. Moreover, the high concentration of SA might improve plant disease resistance by inhibiting FON sporulation [26]. Thereafter, we further demonstrated that the expression of genes related to phytohormones signal pathways showed significant changes at 7 dpi compared to resistant and susceptible watermelon varieties. Our KEGG analysis results identified the 11 up-regulated differently expressed genes most enriched in terpenoid backbone biosynthesis pathways. The GO enrichment classification results indicated that the up-regulated genes were mainly enriched in response to stress and membrane-enclosed lumen, and the *Cla97C04G073730* (clathrin light chain, cellular component) was significantly expressed in two varieties at 7 dpi. These results suggest that the signal transduction of these phytohormones might regulate genes involved with the Clathrin light chain from the plasma membrane into coated vesicles after FON infection, such as *Cla97C04G073730* (Figure 5). The expression of the genes related to phytohormone signal pathways showed significantly changes at 7 dpi compared to resistant and susceptible watermelon varieties. Notably, a number of SA, JA, and ABA signaling component-related genes were identified by transcriptome data, and we further have selected the DEGs encoding the key components involved in the SA, JA, and ABA signaling pathways. Furthermore, our results suggest that the up-regulated expression of the CIPALs and BAH genes was key to SA accumulation in watermelon roots (Figure 6, Figure 7 and Figure 8), and the lower SA content in the resistant variety may help activate watermelon resistance to FON infection at 7 dpi [11,26]. On the contrary, there was a highly elevated SA content in the SF7 group, which was one of factors that led to plant death, and the reason for the SA accumulation might be the disabled metabolisms or some modification activities in the susceptible variety [11,33].

Moreover, increasing evidence suggests that WRKY transcription factors play an essential role in plant defense to pathogen infection [33,34,35]. For instance, WRKY70 has been reported as a mediator of suppression of JA and ABA responses by SA in *Arabidopsis thaliana* [36]. In our study, the significantly expressed WRKYs genes may interfere with SA, JA, and ABA downstream signaling in resistant watermelon plants. Moreover, the diverse expression of four NPRs [37,38,39] of SA receptors, JAR1 [40,41] of JA receptor, and four PYLs [42,43] of ABA receptor genes in watermelon suggest that they may have synergistic roles in regulating watermelon resistance to *Fusarium* wilt by transcription factors, such as WRKYs. Recent studies have reported that there were two groups of SA receptors, NPR1 and NPR3/NPR4. Here, our results indicate that the NPR5 (*Cla97C04G071000*) may play key role in transcriptional regulation of TGA family genes expression. The up-regulated JAR1 (*Cla97C05G081210*) gene in resistant watermelon, which was thought to be involved in activating pathogen-related genes (PR) signal at the onset stage (7 dpi). Moreover, the ABA receptor PYL family genes were essential to regulate its downstream signaling. Additionally, our cis-acting regulatory element of candidate gene promoter’s prediction analysis indicates that the JAR, NPRs, and PYLs family genes in resistant watermelon varieties may trigger plant immune system against FON infection by a crosstalk net between SA, JA, and ABA. For instance, the lower concentration of SA and JA were conducive to plant disease resistance. The ABA content in the resistant group was slightly higher than in the susceptible group at 7 dpi, which implies that the relatively high levels of ABA may promote watermelon growth in the resistant group. In conclusion, our results demonstrate the important role of SA, JA, and ABA in watermelon resistance to FON infection. Furthermore, the results provide evidence for research on watermelon resistance breeding.

## 4. Materials and Methods

### 4.1. Experimental Site Description and Sampling

This study was conducted at Hunan Academy of Agricultural Sciences in the city of Changsha, Hunan Province, China (112°5842 E, 28°1149 N). The soil was sandy loam with background sterilization before separate into each pot (LDZM-80KCS-3 vertical pressure steam sterilizer, ZHONGAN, Shanghai, China) [3]. The trial crops were watermelon resistance cultivated variety PI296341 Zhengzhou Fruit Research Institute, Chinese Academy of Agricultural Sciences (ZFRI, CAAS) and susceptible cultivated variety zaojia 8424 (Xinjiang Farmer Seed Technology Co., Ltd., Urumqi, China). The watermelon seedling nutrition bowl was cultivated and grown in a biochemical incubator (LRH-300, ZHUJIANG, Taihong, Shaoguan, China) at Tm 25 °C, light 16 h/Tm 18 °C, dark 8 h. The nutrition bowl seedling substrate include peat, perlite, and vermiculite (6: 3: 1). We transplanted each plant into pots separately after 30 days.

The pathogenic isolate of *Fusarium* strain FON was firstly incubated in the dark for 7 days on PDA at 28 °C. Then, a bam plug was selected from a PDA plate and placed into 300 mL of potato dextrose broth in a flask, before propagation on a rotary shaker at 200 rpm at 26–30 °C. Detection of FON conidia concentration by blood cell counting plate and adjusted with sterile water to a final concentration of 1 × 10^6^ conidia/mL. When the watermelon seedlings growth was at the stage of two leaves apart, aliquots of 5 mL FON were added into the root zone of each watermelon plant, respectively.

Samples referred to as S, Susceptible cultivar + mock-inoculation control (H_2_O_2_); R, Resistant cultivar + mock-inoculation control (H_2_O_2_); SF, Susceptible cultivar + FON; RF, Resistant cultivar + FON. Moreover, we set five different sampling times before and after FON inoculation, which stopped at 7 days post inoculation as the disease symptoms appeared (yellowing and wilting). Such as 0 dpi (before treatment); 12 hpi (12 h post inoculation); 1 dpi (1 day post inoculation); 3 dpi (3 days post inoculation); 5 dpi (5 days post inoculation); 7 dpi (7 days post inoculation). We selected 10 watermelon plants as one repetition, and set three independent replicates (30 plants) for each sample group, at five different sampling times with every 4 groups, with total of 720 plant samples from 720 separate pots collected.

### 4.2. Determination of the Physiological and Biochemical Indexes in Watermelon Plant

Firstly, we collected each watermelon root with sterilized scissors, for measured the fresh weight of roots with an electric balance, and then counted the number of roots. Then immediately repack the roots with sterilized 5 mL centrifuge tube and stored at−80 °C.

The peroxidase (POD) activity and phenylalanine-ammonialyase (PAL) activity of the plant samples were analyzed respectively, using the BC0095 Peroxidase assay kit and BC0215 PAL test kit (Beijing Solarbio Science & Technology Co., Ltd., Beijing, China), according to manufacturer’s protocols. The malondialdehyde (MDA) content was determined by the thiobarbituric acid method using the BC0025 MDA assay kit (Beijing Solarbio Science & Technology Co., Ltd., Beijing, China), according to manufacturer’s protocols. The Tecan-SPARK Tecan Trading AG, Männedorf, Switzerland) and Eppendorf 5415R refrigerated centrifuge (Eppendorf AG, Hamburg, Germany) was used to test these enzyme activities. Three biological replicates per sample with three technical replicates were performed.

The disease incidence (%) = (No. of infected plants/total number of plants surveyed) × 100%. We selected 10 watermelon plants as one repetition, and set three independent replicates for each sample group, with total of 60 plant samples collected.

### 4.3. Establishment of SA, ABA, and JA Determination System in Watermelon Root

The SA, ABA, and JA contents were measured using LC-MS (Liquid chromatography-tandem mass spectrometry) as per the following process: We selected 10 watermelon plants as one repetition, and set three independent replicates for each sample group, with total of 60 plant samples collected. Every 200 mg fresh watermelon root samples was frozen with liquid nitrogen and homogenized using a Tissue Lyser homogenizer (Qiagen, Germantown, MD, Germany). Thereafter, 1 mL of 80% methanol was added, and the homogenates were mixed in an ultrasonic bath (30 °C) and stored overnight (4 °C). The supernatant was collected (centrifugation at 15,000 g for 10 min) and vacuumed to dry in a Jouan RCT-60 concentrator. Then, the dried extract was dissolved in 200 μL of sodium phosphate solution (0.1 mol L-1, pH 7.8) and passed through a Sep-Pak C18 cartridge (Waters) eluted with 1.5 mL of 80% methanol. After vacuumed to dry again, and eluate was dissolved in 10 mL of 10% methanol, and 5 μL of the solution was injected into the liquid chromatography–tandem mass spectrometry system (LCMS-8030, Shimadzu Corporation, Chiyoda-ku, Japan) [44]. The sample was separated by liquid chromatography and then entered the mass spectrometry. After being ionized by ion source, the ion fragments were separated by mass number by mass analyzer, and the mass spectrum was obtained by detector. Three biological replicates per sample with three technical replicates were performed.

### 4.4. RNA-Seq Sample Collection and Preparation

To examine the molecular mechanism roles of SA, JA, and ABA in resistance to watermelon *Fusarium* wilt, we used transcriptome sequences to comparison analyze the genes expression in different watermelon varieties at the onset stage. The samples are referred to as SF7, Susceptible cultivar + FON, 7 days post inoculation (7 dpi); RF7, Resistant cultivar+FON, 7 days post inoculation (7 dpi). The RNA was extracted from watermelon roots by CTAB standard extraction method. The final RNA concentration and purity were determined using a NanoDrop 2000 UV–Vis spectrophotometer (Thermo Scientific, Wilmington, DE, USA), and the RNA quality was checked by 1% agarose gel electrophoresis (EPS-300, TANON Science & Technology Co., Shanghai, China). The insert size of the library was detected by Agilent 2100 bioanalyzer with an RNA concentration > 200 ng/ul, RNA Integrity Number (RIN) ≥ 8.0, OD260/280 ≥ 1.8, and OD260/230 ≥ 1.5. The RNA-seq transcriptome library was prepared following the TruSeq RNA sample preparation Kit from Illumina (Illumina, San Diego, CA, USA) using 1μg of total RNA. Messenger RNA was isolated according to the polyA selection method by oligo beads and then treated with a fragmentation buffer. Second, double-stranded cDNA was synthesized using a SuperScript double-stranded cDNA synthesis kit (Invitrogen, Inc., Carlsbad, CA, USA) with random hexamer primers (Illumina, San Diego, CA, USA). Then, the synthesized cDNA was subjected to end-repair, phosphorylation, and ‘A’ base addition according to Illumina’s library construction protocol. Libraries were size-selected for cDNA target fragments of 250–300 bp on 2% low range ultra-agarose followed by PCR amplified using Phusion DNA polymerase (NEB) for 15 PCR cycles. After quantified by TBS380, the paired-end RNA-seq sequencing library was sequenced with the Illumina HiSeq platform. Three biological replicates per sample were analyzed. The sequencing was performed at Novogene Biotechnology Co., Ltd. Novogene Co., Ltd. Beijing, China. The clean reads were deposited into the NCBI Sequence Read Archive database (Accession Number: PRJNA794199).

### 4.5. Quantification of Gene Expression Level and Differential Expression Analysis

The reference gene version is watermelon 97103 in the Cucurbit Genomics Database (http://cucurbitgenomics.org/organism/2, accessed on 1 December 2021). Raw data (raw reads) of fastq format were firstly processed through in-house perl scripts. In this step, clean data (clean reads) were obtained by removing reads containing adapter, reads containing ploy-N and low-quality reads from raw data. At the same time, the Q20, Q30, and GC clean data content was calculated. All the downstream analyses were based on the clean data with high quality. Reference genome and gene model annotation files were downloaded from genome website directly. Index of the reference genome was built using Hisat2 v2.0.5. We selected Hisat2 as the mapping tool as Hisat2 can generate a database of splice junctions based on the gene model annotation file and thus a better mapping result than other non-splice mapping tools. The FPKM of each gene was calculated based on the length of the gene and reads count mapped to this gene. Differential expression analysis of two groups (three biological replicates per condition) was performed using the DESeq2R package (1.20.0). DESeq2 provide statistical routines for determining differential expression in digital gene expression data using a model based on the negative binomial distribution. Genes with an adjusted *p*-value < 0.05 found by DESeq2 were assigned as differentially expressed. Principal component analysis (PCA) was used to reduce the dimension of gene variables and extract principal components to evaluate the difference and the duplication of samples between groups. The quantitative distribution of transcripts in each gene set was displayed by Venn diagram.

To identify DEGs (differential expression genes) between different samples, the expression level of each transcript was calculated according to the fragments per kilobase of exon per million mapped reads (FPKM). RSEM was used to quantify gene abundances. R statistical package software EdgeR [45] was utilized for differential expression analysis. The results of differentially expressed genes were displayed by heatmap (https://software.broadinstitute.org/morpheus/, accessed on 1 December 2021). The candidate genes sequences were searched by BLASTp at NCBI. Prediction and analysis of cis-elements in candidate gene promoters and phytohormones at http://bioinformatics.psb.ugent.be/webtools/plantcare/html/ (accessed on 1 December 2021).

### 4.6. Enrichment and Expression Level Analysis of Differential Genes

Gene Ontology (GO) enrichment analysis of differently expressed genes was implemented by the clusterProfiler R package, in which gene length bias was corrected. GO terms with corrected Pvalue less than 0.05 were considered significantly enriched by differential expressed genes. We used clusterProfiler R package to test the statistical enrichment of differential expression genes in KEGG pathways (http://www.genome.jp/kegg/, accessed on 1 December 2021). The 30 most significant terms in the enrichment analysis results were selected to draw the histogram. The size of the dot represents the number of genes annotated. PPI analysis of differently expressed genes was based on the STRING database, which knows and predicts Protein–Protein Interactions. The network was performed by Cytoscape v3.8.2 software.

### 4.7. Quantitative Detection of Candidate Genes by RT-qPCR

The plant roots RNA was extracted using the TRIzol Plant RNA Purification Reagent (Invitrogen, Inc., Carlsbad, CA, USA). A Promega GoScript Reverse Transcription System (Promega Biotech Co., Madison, WI, USA) was used for obtaining cDNA. The final cDNA concentration and purity were determined using a NanoDrop 2000 UV–Vis spectrophotometer (Thermo Scientific, Wilmington, DE, USA), and the cDNA quality was checked by 2% agarose gel electrophoresis (EPS-300, TANON Science & Technology Co., Shanghai, China) with a concentration >150 ng/ul. Distinct regions of candidate rRNA genes were amplified by PCR (Eppendorf AG, Bio-Rad Laboratories, Inc. Hercules, CA, USA) using specific primers (Appendix A). Then the PCR products were used as templates to construct the standard curve of the fluorescence quantitative PCR (Bio-Rad CFX96 Real-Time System, Bio-Rad Laboratories, Inc. Hercules, CA, USA). The correlation coefficients (R2) of other candidate genes were more than 0.98 and the PCR efficiency between 90 and 110%. The expression levels were calculated using the 2-ΔCT method. Primer Premier 5.0 (Premier, Inc., CAN, Charlotte, NC, USA) was used to design family specific primers. For the sequences with high homology, dnaman 7.0 (Lynnon Biosoft, San Ramon, CA, USA) was used for multiple sequence alignment, and primers were designed in the non-conservative region. CIACT was used as reference genes [10].

### 4.8. Statistical Analysis

Statistical analysis was performed using GraphPad Prism 9 (GraphPad Software, San Diego, CA, USA). Differences between two treatments were tested by multiple t tests of Two-way ANOVA. All the values were expressed as mean ± standard error. The figures were performed by using Microsoft Office 2010 (Microsoft Corporation, Redmond, WA, USA).

## 5. Conclusions

In this experiment, we have established the SA, JA, and ABA determination system in watermelon roots. Through sequencing analysis and RT-qPCR, we have identified the significantly expressed genes involved in SA, JA, and ABA pathways, which play an important role in activating watermelon defense responsibility to against FON. Moreover, these differentially expressed NPRs, JAR, and PYLs family genes in resistant watermelon varieties may play a crosstalk role in signal transport between SA, JA, and ABA to activate the watermelon plant immune system against FON infection. Overall, our data provided a comprehensive resource for identify the genes associated with phytohormones of watermelon resistance breeding.

## Figures and Tables

**Figure 1 plants-11-00156-f001:**
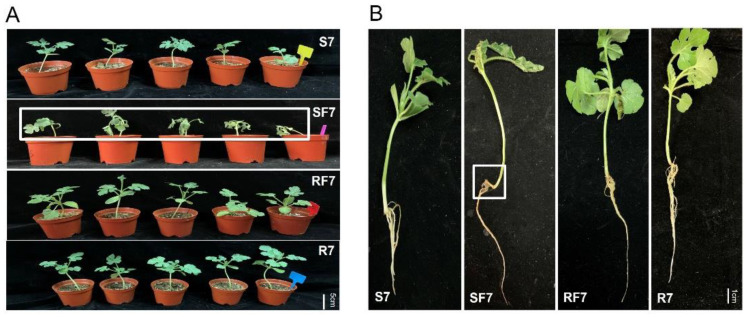
Comparison analysis of watermelon performance after FON infection at 7 dpi: (**A**) Comparison of watermelon seeding phenotype in different samples; (**B**) Comparison analysis of root phenotype in different samples.

**Figure 2 plants-11-00156-f002:**
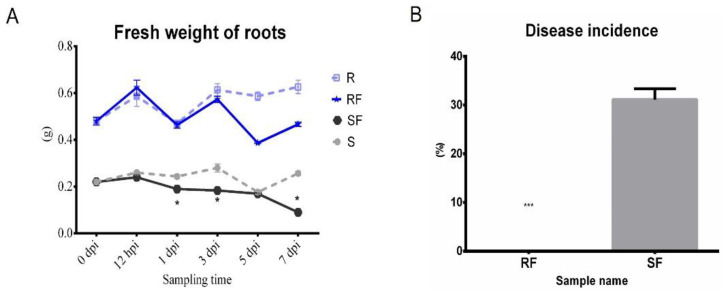
Comparison analysis fresh weight of roots and *Fusarium* wilt disease occurrence on different watermelon varieties after FON inoculation: (**A**) Comparison analysis of fresh weight of roots in different samples; (**B**) Comparison analysis of disease incidence in different samples. S, Susceptible cultivar + mock-inoculation control (H_2_O_2_); R, Resistant cultivar + mock-inoculation control (H_2_O_2_); SF, Susceptible cultivar + FON; RF, Resistant cultivar + FON. Moreover, 0 dpi (before treatment); 12 hpi (12 h post inoculation); 1 dpi (1 day post inoculation); 3 dpi (3 days post inoculation); 5 dpi (5 days post inoculation); 7 dpi (7 days post inoculation). Data were expressed as mean ± SE (*n* = 3). Multiple t tests of Two-way ANOVA (*, *p* ≤ 0.05; ***, *p* ≤ 0.0001).

**Figure 3 plants-11-00156-f003:**
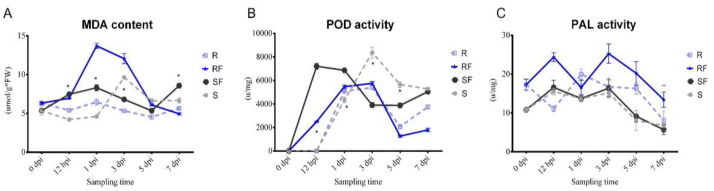
Comparison analysis of physiological indexes changes: (**A**) Dynamic changes in MDA content at different sampling time in different samples; (**B**) Dynamic changes in POD enzyme activities at different sampling time in different samples; (**C**) Dynamic changes in PAL enzyme activities at different sampling time in different samples. S, Susceptible cultivar + mock-inoculation control (H_2_O_2_); R, Resistant cultivar + mock-inoculation control (H_2_O_2_); SF, Susceptible cultivar + FON; RF, Resistant cultivar + FON. Moreover, 0 dpi (before treatment); 12 hpi (12 h post inoculation); 1 dpi (1 day post inoculation); 3 dpi (3 days post inoculation); 5 dpi (5 days post inoculation); 7 dpi (7 days post inoculation). Data were expressed as mean ± SE (*n* = 3). Multiple *t* tests of Two-way ANOVA (*, *p* ≤ 0.001; **, *p* ≤ 0.0001).

**Figure 4 plants-11-00156-f004:**
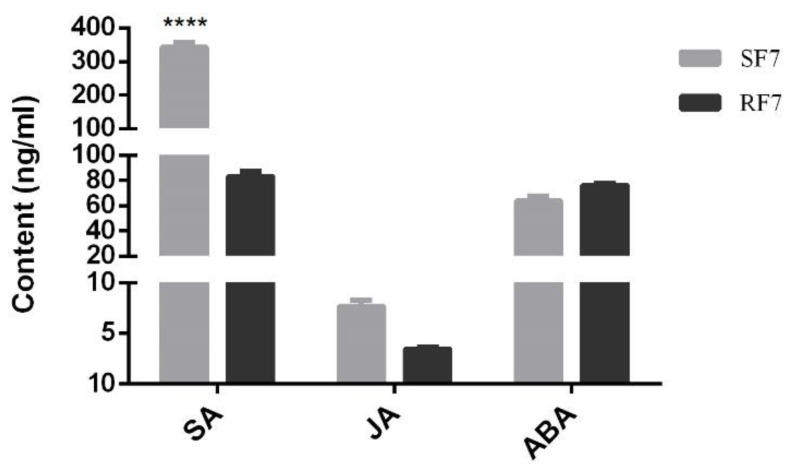
Comparison analysis of SA, JA, and ABA contents in different samples. SF7, Susceptible cultivar + FON, 7 days post inoculation (7 dpi); RF7, Resistant cultivar + FON, 7 days post inoculation (7 dpi). Data were expressed as mean ± SE (*n* = 3). Multiple t-test of Two-way ANOVA (****, *p* ≤ 0.0001).

**Figure 5 plants-11-00156-f005:**
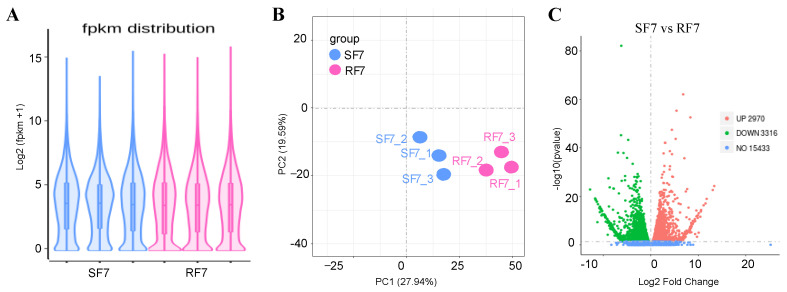
RNA-Seq comparison analysis of watermelon gene expression in resistance and sensitivity varieties: (**A**) The violin diagram of FPKM distribution; (**B**) PCA analysis of different samples; (**C**) Volcano map of DEGs between resistant and susceptible cultivars. Note: SF7, Susceptible cultivar + FON, 7 days post inoculation (7 dpi); RF7, Resistant cultivar + FON, 7 days post inoculation (7 dpi). Three independent replicates.

**Figure 6 plants-11-00156-f006:**
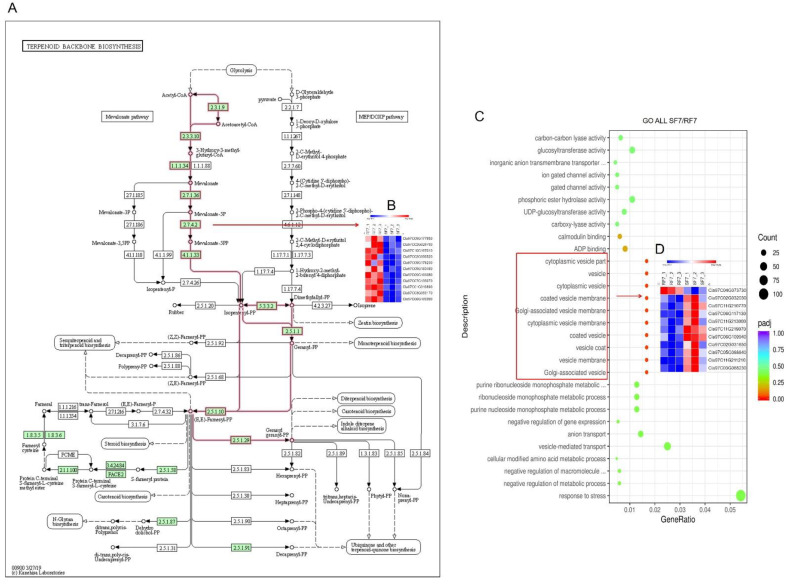
Function enrichment analysis of differential watermelon varieties: (**A**) Different expressed genes in Terpenoid backbone biosynthesis pathways; (**B**) Heatmap of 10 DEGs in Terpenoid backbone biosynthesis pathways; (**C**) All genes GO enrichment significant analysis; (**D**) Heatmap of 11 DEGs in GO enrichment analysis. Note: SF7, Susceptible cultivar +FON, 7 days post inoculation (7 dpi); RF7, Resistant cultivar+FON, 7 days post inoculation (7 dpi). Three independent replicates. Blue bands indicate low gene expression and red bands high gene expression.

**Figure 7 plants-11-00156-f007:**
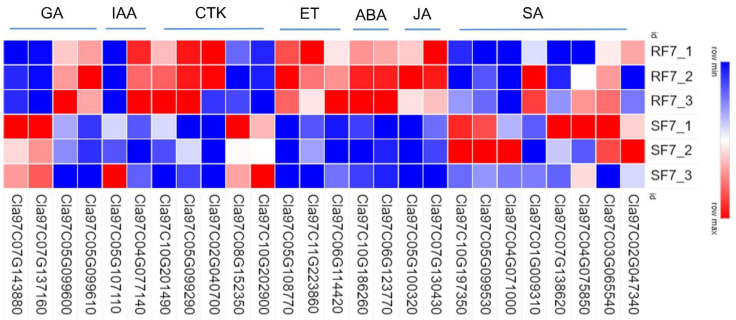
Comparison analysis of relative expressions of candidate genes related to phytohormone pathways in resistant and susceptible samples: GA: gibberellins; IAA: auxin; CTK: cytokinin; ET: ethylene; ABA: abscisic acid; JA: jasmonic acid; SA: salicylic acid. Note: SF7, Susceptible cultivar +FON, 7 dpi; RF7, Resistant cultivar + FON, 7 dpi. Three independent replicates.

**Figure 8 plants-11-00156-f008:**
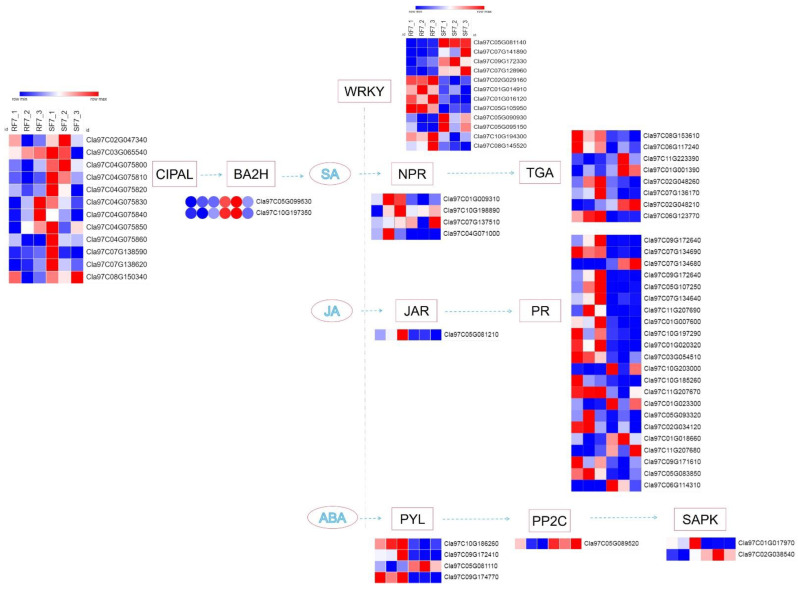
The schematic of SA, JA, and ABA signaling-related genes network in watermelon roots at 7 days after FON infection: Note: SF7, Susceptible cultivar + FON, 7 days post inoculation (7 dpi); RF7, Resistant cultivar + FON, 7 days post inoculation (7 dpi). Three independent replicates. Blue bands indicate low gene expression and red bands high gene expression.

**Figure 9 plants-11-00156-f009:**
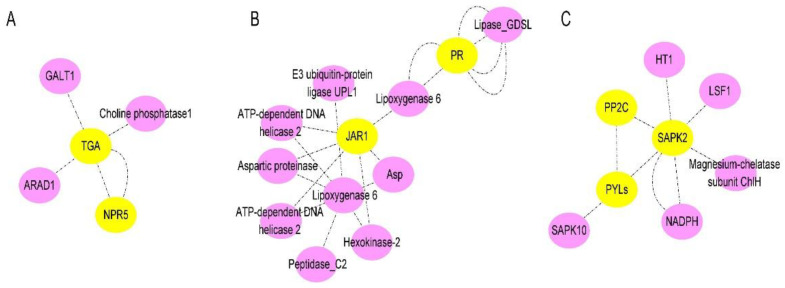
The schematic of SA, JA, and ABA signaling-related proteins network in watermelon: (**A**) Interaction network of SA signaling-related proteins in watermelon roots at 7 days after FON infection; (**B**) Interaction network of JA signaling-related proteins watermelon roots at 7 days after FON infection; (**C**) Interaction network of ABA signaling-related proteins watermelon roots at 7 days after FON infection. Each node in the diagram represents a protein, and each connecting line represents the interaction between connected proteins.

**Figure 10 plants-11-00156-f010:**
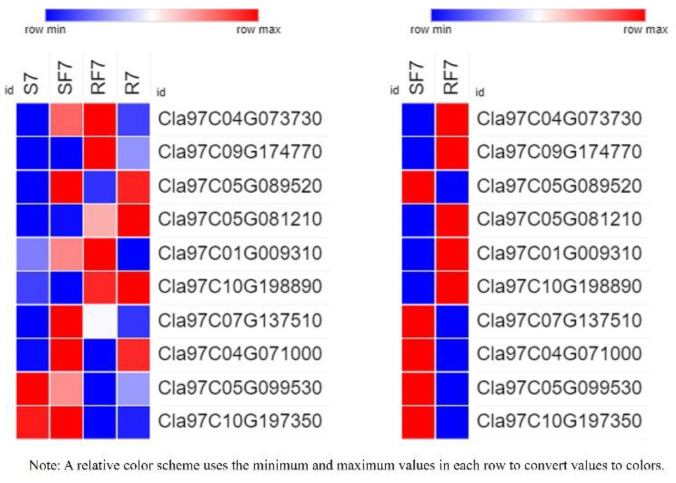
Relative expressions of 10 candidate genes in different samples by RT-qPCR. Note: S7, Susceptible cultivar + mock-inoculation control (H_2_O_2_), 7 dpi (7 days post inoculation); R7, Resistant cultivar + mock-inoculation control (H_2_O_2_), 7 dpi (7 days post inoculation); SF7, Susceptible cultivar + FON, 7 dpi (7 days post inoculation); RF7, Resistant cultivar + FON, 7 dpi (7 days post inoculation). Three biological replicates per samples were analyzed.

## Data Availability

The clean reads were deposited into the NCBI Sequence Read Archive (SRA) database (Accession Number: PRJNA794199).

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
