# Peer review of "Study on the Role of Phytohormones in Resistance to Watermelon *Fusarium* Wilt"

_plants, 2022, doi:10.3390/plants11020156_

Round 1

Reviewer 1 Report

Dear colleagues. Please pay attention to the comments and suggestions in the attached file.

Author Response

Response to Reviewer 1 Comments

Point 1: The phrase on page 2, line 70-71 should be changed somehow. “Furthermore, the disease incidence of watermelon Fusarium wilt in SF5 was 33.3%, - while was 0% in RF5 (Fig. 2B), which confirm that the resistant variety have a resistance to FON infection”. The fact that resistant varieties are resistant is obvious and word infection is superfluous.

Response 1: Thanks for your kindly suggestion. We have deleted the “which confirm that the resistant variety have a resistance to FON infection” in line 87.

Point 2: As for Figure 2, page 3, line 81-84

I propose to change the designations. For example, to designate uninfected variants with empty symbols and a dotted line, infected variants with filled symbols and a solid line.

Response 2: Thanks for your kindly suggestion. We have changed the designations in Figure 2. For example, to designate uninfected variants with empty symbols and a dotted line, infected variants with filled symbols and a solid line. Please see the revised Figure2 and Figure 3 in line 94, 119.

Point 3: Figure 2 В, specify the number of plants used in the samples and the number of samples.

Response 3: Thanks for your kindly suggestion. We have specified the number of plants used in the samples and the number of samples in line 375-378.

Point 4: Page 3, line 90-93 «Moreover, 0 denotes before treatment; 190 denotes 12hpi (12 hours post inoculation); 2 denotes 1dpi (1 day post inoculation); 3 denotes 3dpi91 (3 days post inoculation); 4 denotes 5dpi (5 days post inoculation); 5 denotes 7dpi (7 days post92 inoculation)». Such a designation seems too complicated. It is better to directly indicate the number of days by indexes, 0,5; 1; 3; 5; 7. R5 instead of 7 is confusing.

Response 4: Thanks for your kindly suggestion. We have revised all the sample names in the manuscript according to your suggestion. Please see line 76-79 and other related with highlights in the manus.

Point 5: "Page 3, line 103-104 “Notably, the POD enzyme activity in SF group was nearly 2 times higher than that in RF group at 3dpi (Fig. 2D)”. What does "group" mean and where?

Response 5: Thanks for your kindly suggestion. We have changed the “SF group” to “susceptible cultivar” in line 85.

Point 6: Page 4, line 126-129. “Our data indicate that the SA was induced after FON attack in both varieties of watermelon roots, but had a higher enrichment in susceptible group at onset stage.The results confirm that the low concentration of SA is conducive to plant disease resistance while excessive accumulation of SA may lead to plant death [11,13]”. Are there references to similar studies? More often, the high concentration of salicylic acid associated with resistance.

Response 6: Thanks for your kindly suggestion. We have moved these sentences in discussion section and added a sentence with reference to explain why the high concentration of SA associated with resistance. Please see line 302-304.

Point 7:  "Page 6, line 149. Figure 5. The pictures are very illegible.

Response 7: Thanks for your kindly suggestion. We have revised the pictures in new Figure 6. Please see line 186.

Point 8: Page 11, line 325-326. “For instance, the lower concentration of SA and JA were conducive to plant disease resistance”. See the note to the page line 90-93

Response 8: Thanks for your kindly suggestion. We have revised all the sample names in the manuscript according to your suggestion. Please see line 369-375.

Point 9: Page 11, line 352-356. «Moreover…». See the note to the page 3, line 90-93.

Response 9: Thanks for your kindly suggestion. We have revised all the sample names in the manuscript according to your suggestion. Please see line 369-375.

Point 10: Page 12, line 372-373. What was the sample size when determining symptoms? Was any scale used?

Response 10: Thanks for your kindly suggestion. We have added sample information in line 395-396.

Point 11: Page 14, line 478 “Diff erences” there is a typo.

Response 11: Thanks for your kindly suggestion. We have deleted the typo in “Diff erences” in line 504.

Point 12: Statistical analysis was performed using GraphPad Prism 9 (GraphPad Software, San Diego, CA, USA). Diff erences between two treatments were tested by independent sample t-test at P ≤ 0.05” Was the compliance of the sample with the normal distribution

checked before applying the t-test?

Response 12: Thanks for your kindly suggestion. We have rewritten the sentence in line 504-505.

Reviewer 2 Report

The manuscript plants-1514678 studied the salicylic acid, jasmonic acid, and abscisic acid phytohormones in watermelon roots and analyzed their roles against Fusarium wilt. The authors compared resistant and susceptible varieties using transcriptome sequencing and RT-qPCR.

The manuscript topic is of great interest, the methods applied appropriate, and the findings reported considerable. However, the manuscript should be ameliorated for some major issues mainly related to the quality of presentation of some sections.

Broad comments

  • Abstract: Authors should avoid the use of headings and undetailed abbreviations.
  • Introduction: The introduction does not place the study in a broad context, introducing the importance of the work. The authors should also clearly define the working hypothesis. 
  • Results: The results description should be ameliorated. For specific comments see the pdf file attached. 
  • Discussion: The authors correctly discussed the results from the perspective of previous studies and the purpose of the study. I would add more details on the importance of the findings obtained. Which are the implications of your findings?
  • Materials and Methods: The authors should describe in more detail some of the methods used.

Supplementary material was not available for consultation.

Specific comments are provided in the pdf file.

The English language should be checked for grammar and typos.

Author Response

Dear reviewer,

Thanks for all your kindly suggestion. We have made major revisions following all the suggested revisions involve minor changes (especially introduction, results, discussion, and materials & methods parts, and the grammar check of English language). The point by point, the details of the revisions to the manuscript and our responses to the referees’ comments are in the attachment file.

Best regards,

Feiying Zhu

Round 2

Reviewer 2 Report

The authors correctly addressed all my previous comments. I have no further suggestions.